# Risk Evaluation of Ice Flood Disaster in the Upper Heilongjiang River Based on Catastrophe Theory

Yu Li [1], Hongwei Han [1,2,*], Yonghe Sun [3], Xingtao Xiao [3], Houchu Liao [3], Xingchao Liu [1] and Enliang Wang [1,2]

1　School of Water Conservancy and Civil Engineering, Northeast Agricultural University, Harbin 150030, China; 18846771809@163.com (Y.L.); dnlxc@neau.edu.cn (X.L.); hljwel@126.com (E.W.)
2　Heilongjiang Provincial Key Laboratory of Water Resources and Water Conservancy Engineering in Cold Region, Northeast Agricultural University, Harbin 150030, China
3　Heilongjiang Provincial Hydrology and Water Resources Center, Harbin 150001, China
*　Correspondence: hanhongwei@neau.edu.cn

**Abstract:** The ice flood phenomenon frequently occurs in frigid locations of high latitude and high altitude, which triggers ice dam or ice jam flooding thus endangering personal and property safety. Hence, a scientific risk evaluation with enough consideration of each factor is a basic and necessary requirement for preventing ice flood disaster risks. This study establishes a risk evaluation system for ice flood disasters based on the catastrophe theory and utilizes the Pearson correlation coefficient to screen underlying indicators to evaluate the risk of ice flood in the upper Heilongjiang River region. Considering the correlation between different indicators, a hierarchical cluster analysis is invoked to simplify the indicator set and to select typical years. The results of the evaluation system indicate that the catastrophe membership values in the Mohe, Tahe, and Huma regions from 2000 to 2020 ranged from 0.86 to 0.93. Based on the membership values and the actual disaster situations, a four-level classification of risk ratings is conducted. The comparison between the results obtained from the catastrophe theory evaluation method and the fuzzy comprehensive evaluation method reveals similar risk levels, which verifies the effectiveness and practicality of the catastrophe theory applied to the ice flood risk evaluation and presents a novel method for the study of ice floods.

**Keywords:** ice flood disaster; catastrophe theory; indicator preference; risk evaluation; Heilongjiang River

## 1. Introduction

Ice flooding is a unique hydrological phenomenon that occurs in frigid region rivers. It is mainly manifested in the flow and evolution of river ice during the winter and spring seasons, and due to a reduced overflow cross-section, ice jams and dams are formed, resulting in backwater staging and a high upstream water level, which can rapidly create a flooding hazard at short notice [1,2]. Due to the high backwater levels, extremely fast ice flow rates, and complex formation mechanisms of ice-induced floods, showing the relationship between floodwater levels and the probability of occurrence in hazardous watersheds becomes more difficult [3]. Therefore, it is significantly important to ensure that the risk evaluation of ice flood disasters is correct and has enough safety margin in regional disaster prevention and mitigation capabilities. Ice-jam floods often cause issues such as farmland submersion, building collapse, and embankment erosion. Additionally, these disasters can also affect the sedimentation and release of chemical substances in river water bodies [4,5]. Thus, ice floods are more likely than open-water floods to cause financial costs and damage to humans and their habitats [6]. In an ice flood risk evaluation, both the probability of occurrence, commonly quantified as the return period, and the damage consequences of the flood hazard need to be considered to assess the annual expected costs [7,8]. This is not only a necessary prerequisite for the prevention and control of natural disasters in frigid regions but also a crucial assurance for national sustainable development and the effective execution of significant programs.

Numerous elements, including environmental changes, water flow conditions, river borders, human activities, and socioeconomics, impact the evaluation of ice flood risk [8–13]. Due to its complicated hazard-inducing environment, various hazard factors, and fragile hazard-bearing body, it has been a challenging and popular subject of disaster research. While open-water flood hazard delineation and risk analysis are commonly used in traditional flood management approaches, methodologies for ice-related flood hazard delineation and risk analysis are not yet well established. This is due to the highly unpredictable and complex nature of ice-related flood events, which present significant challenges for accurate evaluation and assessment [14]. For this reason, some scholars have conducted a lot of research on the study of causes of ice floods, the forecasting of ice-jam and ice-dam floods, the delineation of flooding hazards, and flood risk calculation methods. The formation and evolution of ice floods are extremely complicated, and most scholars are currently studying the impact of various factors on the causes and disasters of flooding according to the geographical environment, hydrology, meteorology, and engineering of rivers [15]. The hydrotechnical approach proposed is the most appropriate method for assessing the risk of ice-jam floods in river systems when adequate historical and on-site data related to ice blockage are available [16]. However, the reliability of hydrometric gauge data can be compromised due to the extreme forces exerted by ice debris and blocks, which may lead to the unavailability and inaccuracy of the data [17,18]. Meanwhile, due to the harsh regional environments where severe ice floods occur, it is generally impossible to acquire real-time, complete, and precise data from field observations [19]. Beltaos [20] used the distributed-function method (DFM) to determine the frequencies and probabilities of ice-jam floods. Several studies [21,22] have utilized an ice-jam numerical model (RIVICE) to evaluate real-time ice-jam flood hazards along the Athabasca River at Fort McMurray, etc. These studies can establish a basis for real-time ice-jam flood risk analysis and improve our comprehension of the ice-jam flood risk of both property and inhabitants. Artificial intelligence techniques, such as neural networks and fuzzy logic systems [13,23], show promise in modeling the nonlinear processes underlying the formation of ice floods. In particular, combining multiple models to predict backwater levels can improve accuracy by 20–30%, albeit at a significant computational cost. Mahabir et al. [24–26] forecasted breakup water levels using multiple linear regression and based on this evaluated the application of soft computing in modeling the maximum water level during river breakout in flood and non-flood event years through fuzzy logic and artificial neural networks. Through the utilization of projection tracing, fuzzy clustering, and accelerated genetic algorithm, Wu [27] created a comprehensive evaluation model of the ice disaster risk that occurred in the Ning-Meng portion of the Yellow River. Luo [28] proposed the GM (1,1) evaluation model, which introduces three-parameter interval gray numbers, to simulate and anticipate the development trend of risk vectors. Numerous studies have already been conducted on flood hazard delineation and risk assessment in the context of managing the risk of ice floods [8,29–31].

In general, the process of nonlinear changes in ice-flooding behavior is a dynamic and irreversible evolutionary process under the influence of the realistic environment. Therefore, the ice flood disruption can be seen as an abrupt state catastrophe phenomenon driven by the energy of the river ice system. This study focuses on the disaster risk evaluation of ice floods using the catastrophe theory. Hazard-inducing environment, hazard factor, hazard-bearing body, and anti-icing capability are considered the criterion layer of the evaluation system, in order to research the indicators in the process of ice flood damage. The catastrophe evaluation method is combined with the Pearson correlation coefficient and hierarchical cluster analysis to solve the problem of index selection and optimization in the evaluation model. The actual ice flood situations in Mohe, Tahe, and Huma regions are taken as an example and are used in establishing the classification of the risk level of the ice flood in the regions. Meanwhile, a comparison is made between the results of the catastrophe theory evaluation method and the fuzzy comprehensive evaluation method,

which verifies the efficacy and applicability of the catastrophe theory applied to the ice flood risk evaluation and introduces a novel concept for the study of ice floods.

## 2. Materials and Methods

### 2.1. Catastrophe Evaluation Method

Calculus, as a mathematical tool, is well-suited for modeling and problem-solving in natural phenomena characterized by continuity and differentiability. It enables the study of continuous, gradual, and smooth changes, allowing for a deeper understanding of such processes in nature. However, when a continuous development undergoes a transition from gradual and quantitative changes to abrupt and qualitative changes, calculus becomes inadequate for describing and addressing such phenomena. In order to solve the step change process, René Thom, a French mathematician, initially introduced the catastrophe theory in 1974 [32] to explore and research discontinuous changes and abrupt variations in phenomena. He discussed the three basic forms of the system and the mathematical principles of structural stability, singularity, and topology and described the transition from continuous asymptotic and quantitative changes to discontinuous jump mutations and qualitative changes using mathematical methods. The catastrophe theory utilizes potential functions to classify the critical points of a system and investigates the characteristics of discontinuous changes near each critical point. The properties of the discontinuous state that are located around the crucial points are uncovered in order to conduct a more in-depth investigation of the process underlying discontinuous occurrences [33]. There are seven different major forms of catastrophe theory [34], as determined by the geometry of the restriction criteria. Among them, different types of primary catastrophe models are used which include the cusp catastrophe type, swallowtail catastrophe type, butterfly catastrophe type, etc. The equilibrium surfaces and singular point sets associated with these models are shown in Table 1 and Figure 1.

**Table 1.** Normalization formulae for the catastrophe theory.

| Category | Potential Function | Normalization Formula |
|---|---|---|
| Cusp | $V(x) = x^4 + ax^2 + bx$ | $x_a = a^{1/2}, x_b = b^{1/3}$ |
| Swallowtail | $V(x) = x^5 + ax^3 + bx^2 + cx$ | $x_a = a^{1/2}, x_b = b^{1/3}, x_c = c^{1/4}$ |
| Butterfly | $V(x) = x^6 + ax^4 + bx^3 + cx^2 + dx$ | $x_a = a^{1/2}, x_b = b^{1/3}, x_c = c^{1/4}, x_d = d^{1/5}$ |

Note: The control variables $a$, $b$, $c$, and $d$ can be viewed as factors influencing the system's behavior state and are decreasing in importance from $a$ to $d$.

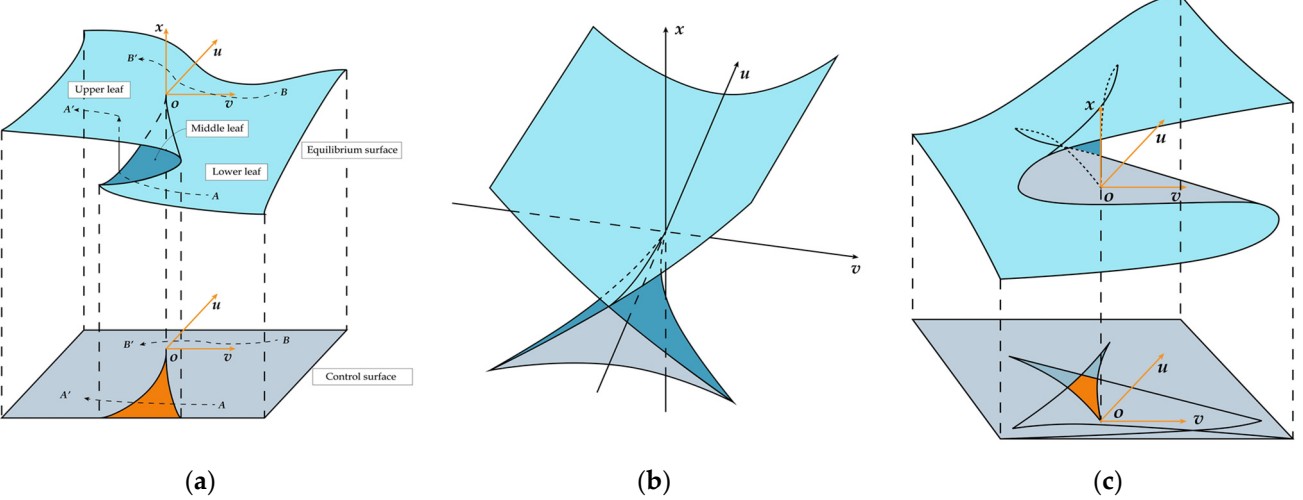

(a)      (b)      (c)

**Figure 1.** Equilibrium surfaces and singular point sets of catastrophe models. (**a**) Cusp catastrophe type; (**b**) Swallowtail catastrophe type; and (**c**) Butterfly catastrophe type.

It is worth noting that due to the four-dimensional and five-dimensional potential functions of the swallowtail and butterfly catastrophe types, the equilibrium surfaces in Figure 1b,c represent the projection of the original functions in three-dimensional space after constraining the variables.

In the process of risk assessment and judgment, it is important to consider the characteristics of each indicator layer and the actual situation. The choice should be made among the following three evaluation principles:

1.  Non-complementary criterion.

In cases where the control variables of the system cannot be substituted for one another, the minimum value among the corresponding mutation values of the $m$ control variables $(a, b, \ldots, m)$ is selected as $x$, as follows:

$$x = min(x_a, x_b, \ldots, x_m) \tag{1}$$

2.  Complementary criterion.

If the variables of a system are mutually substitutable, the corresponding mutation values of each indicator should be calculated according to the catastrophe model of the system. Then, the average value of the variables should be computed as follows:

$$x = \frac{(x_a + x_b + \ldots + x_m)}{m} \tag{2}$$

3.  Over-threshold complementary criterion.

The over-threshold complementarity criterion is based on the analysis of extreme events that exceed specific thresholds. It is used to estimate the tail probability or frequency of events surpassing the threshold. These tail probabilities or frequencies are typically low and are thus considered extreme events or exceptional circumstances. By analyzing these extreme events, we can gain a better understanding of the risk characteristics of the system and take appropriate measures for risk assessment and decision-making. The control variables of the system must reach a certain threshold before they can complement each other [33].

*2.2. Data Preprocessing*

The extreme value method is employed to standardize the preliminary selection indicators, aiming to mitigate the impact of diverse indicator data and their magnitudes on the indicator screening process and to enable effective comparison between indicators. Equation (3) is applicable in situations where larger values of the indicator are more advantageous for analysis. Equation (4) is applicable in situations where smaller values of the indicator are more advantageous for analysis. Assuming that the system indicators can be described by the state variables $x_{ij}$, the following extreme value normalization transformation formula can be used:

$$y_{ij} = \frac{x_{ij} - \min_{1 \leq i \leq m} \{x_{ij}\}}{\max_{1 \leq i \leq m} \{x_{ij}\} - \min_{1 \leq i \leq m} \{x_{ij}\}} \tag{3}$$

$$y_{ij} = \frac{\max_{1 \leq i \leq m} \{x_{ij}\} - x_{ij}}{\max_{1 \leq i \leq m} \{x_{ij}\} - \min_{1 \leq i \leq m} \{x_{ij}\}} \tag{4}$$

where $y_{ij}$ represents the normalized value of the state variable $x_{ij}$, $\min_{1 \leq i \leq m} \{x_{ij}\}$ represents the minimum value of $x_{ij}$, and $\max_{1 \leq i \leq m} \{x_{ij}\}$ represents the maximum value of $x_{ij}$, $i = 1, 2, 3, \ldots, m$ ($m$ is the designation of years); $j = 1, 2, 3, \ldots, n$ ($n$ is the designation of indicators). Then $y_{ij}$ is the dimensionless data and between 0 and 1.

### 2.3. Pearson Correlation Coefficient

The Pearson correlation coefficient is a statistical measure that quantifies the strength and direction of the linear relationship between two continuous variables and is widely used in various fields of research and data analysis.

- STEP 1: Determine the constraints of Pearson correlation:
    - There is a linear relationship between the two variables;
    - The variables are continuous variables;
    - The variables are normally distributed, and the binary distribution is also normally distributed;
    - The two variables are independent.
- STEP 2: Calculate the Pearson correlation coefficient between $X_i$ and $Y_i$. The Pearson correlation coefficient is represented by the symbol "$r$" and takes values between $-1$ and 1. The coefficient is calculated based on the covariance between the two variables and the product of their standard deviations. The formulation of the correlation coefficient can be described as follows:

$$r = \frac{\sum (X_i - \overline{X})(Y_i - \overline{Y})}{\sqrt{\sum_{i=1}^{n} (X_i - \overline{X})^2 \sum_{i=1}^{n} (Y_i - \overline{Y})^2}} \tag{5}$$

where $X_i$ and $Y_i$ are the individual data points in the two variables, $\overline{X}$ and $\overline{Y}$ are the means of the two variables, and $\Sigma$ denotes summation over all data points.

The resulting value of "$r$" indicates the strength and direction of the linear relationship between the variables:

- If "$r$" is close to 1, it indicates a strong positive linear relationship, meaning that as one variable increases, the other variable also tends to increase;
- If "$r$" is close to $-1$, it indicates a strong negative linear relationship, meaning that as one variable increases, the other variable tends to decrease;
- If "$r$" is close to 0, it indicates a weak or no linear relationship between the variables.

It is noteworthy that in correlation analysis, we typically aim to determine whether the observed correlation coefficient is significantly different from zero, indicating the statistical significance of the correlation relationship. These significance tests are designed to evaluate whether the observed correlation coefficient is sufficiently large to reject the presence of correlation due to random sampling errors [35]. During significance testing, it is necessary to choose an appropriate significance level (typically 0.05) to determine whether to reject the null hypothesis. If the null hypothesis is rejected, it can be concluded that the observed correlation is significant.

### 2.4. Hierarchical Cluster Analysis

Cluster analysis is commonly used in scientific research to identify group associations and assess the affinity among different variables [36–39]. Hierarchical cluster analysis is a data analysis technique used to identify groups or clusters within a dataset based on their similarity or proximity. It is a form of unsupervised learning, as it does not rely on predefined class labels or target variables. Hierarchical cluster analysis, specifically, is used to determine associations between different parameters and ultimately identify the sources and processes related to them [40].

Ice flood disasters encompass various parameters from different disciplines, and there exists a significant correlation among these parameters. This correlation leads to a substantial increase in the time and computational resources required for data collection and processing.

In this study, the hierarchical cluster analysis method is employed to demonstrate the interrelationships among the variables under investigation [41]. Based on the criterion of the sum of distances, the typical years are selected by identifying the cluster centroids.

This approach guarantees the comprehensiveness of the evaluation index system while mitigating potential issues, such as result distortion caused by redundant indicators.

### 2.5. Fuzzy Comprehensive Evaluation Method

The fuzzy comprehensive evaluation method is a mathematical approach used to assess complex systems or phenomena that involve uncertainties and imprecise information [42,43], including the following steps:

- STEP 1: Assuming that there are $n$ years to be evaluated to form a sample set, and based on the eigenvalues of m indicators, the eigenvalue matrix of ice flood risk to be evaluated can be expressed as Equation (6):

$$X = (x_{ij}) = \begin{bmatrix} x_{11} & x_{12} & \cdots & x_{1n} \\ x_{21} & x_{22} & \cdots & x_{2n} \\ \vdots & \vdots & \ddots & \vdots \\ x_{m1} & x_{m2} & \cdots & x_{mn} \end{bmatrix} \tag{6}$$

Upon applying data normalization using Equations (3) and (4), we derived the relative membership matrix $R = (r_{ij})$.

- STEP 2: Construct the index weight set.

In order to account for the varying importance of different factors in evaluating the objective, it is necessary to establish a set of indicator weights. Weighting techniques fall into two primary categories: statistical-based methods and participatory-based methods. The statistical-based methods analyze the indicator data to determine the weights, whereas the participatory-based methods involve incorporating expert or public opinions to determine the weights. In this study, the weight is determined using the entropy weight method, which falls under the statistical-based methods. The index weight set is constructed according to Equation (7).

$$A = (w_1, w_2, \ldots, w_m) \tag{7}$$

- STEP 3: Establishing the fuzzy comprehensive evaluation model.

To obtain the result of the fuzzy comprehensive evaluation for each sample, the evaluation matrix and the index weight set are quantified as shown in Equation (8). The synthesis operators used for fuzzy synthesis calculations include the dominant factor determining operator, the dominant factor prominent operator, and the weighted average operator, among others. Due to the interplay of factors in ice flood risk, this study employs the weighted average operator for fuzzy synthesis calculations.

$$B = A \circ R = (b_1, b_2, \ldots, b_n) \tag{8}$$

where $B$ is the fuzzy comprehensive evaluation risk membership matrix of the assessed object.

## 3. Ice Flood Risk Evaluation

### 3.1. Study Area Overview

The main stream of the Heilongjiang River is located in the high latitude and cold region of the border between China and Russia. The river freezes for a long time and is prone to ice dam and jam disasters during the open river flow period. The Heilongjiang River originates from the Erguna River in the Mongolian Plateau, with a total length of approximately 4440 km and a drainage area of about 1,855,000 square kilometers. The Heilongjiang River flows through various terrains, including mountains, canyons, and plains. Within China's territory, the riverbed is rugged, forming a series of rapids and waterfalls. Upon entering Russia, the river becomes gentle and flows through vast plains.

This study was conducted along a specific section of the Heilongjiang River, spanning approximately 800 km from the Mohe region to the Huma region (Figure 2). The three regions of Mohe, Tahe, and Huma, situated between 50.9° N and 53.5° N, have histori-

cally experienced the highest occurrence of spring ice flood disasters and have been the most severely affected regions [44]. These three regions are part of the Greater Khingan Mountains region, characterized by a cold temperate continental monsoon climate. The average temperature in the region has been recorded at −2.1 °C over the years, while the annual average precipitation remains around 460 mm. Between 2000 and 2020, there were seven years marked by severe ice flood disasters, with no fixed occurrence location. The average duration of these events ranged from 2 to 3 days, while the longest recorded event lasted for 15 days. In the springs of 2000 and 2009, the upper sections of the Heilongjiang River experienced over seven ice dams and jams, resulting in the highest backwater heads reaching 7.58 and 9.23 m.

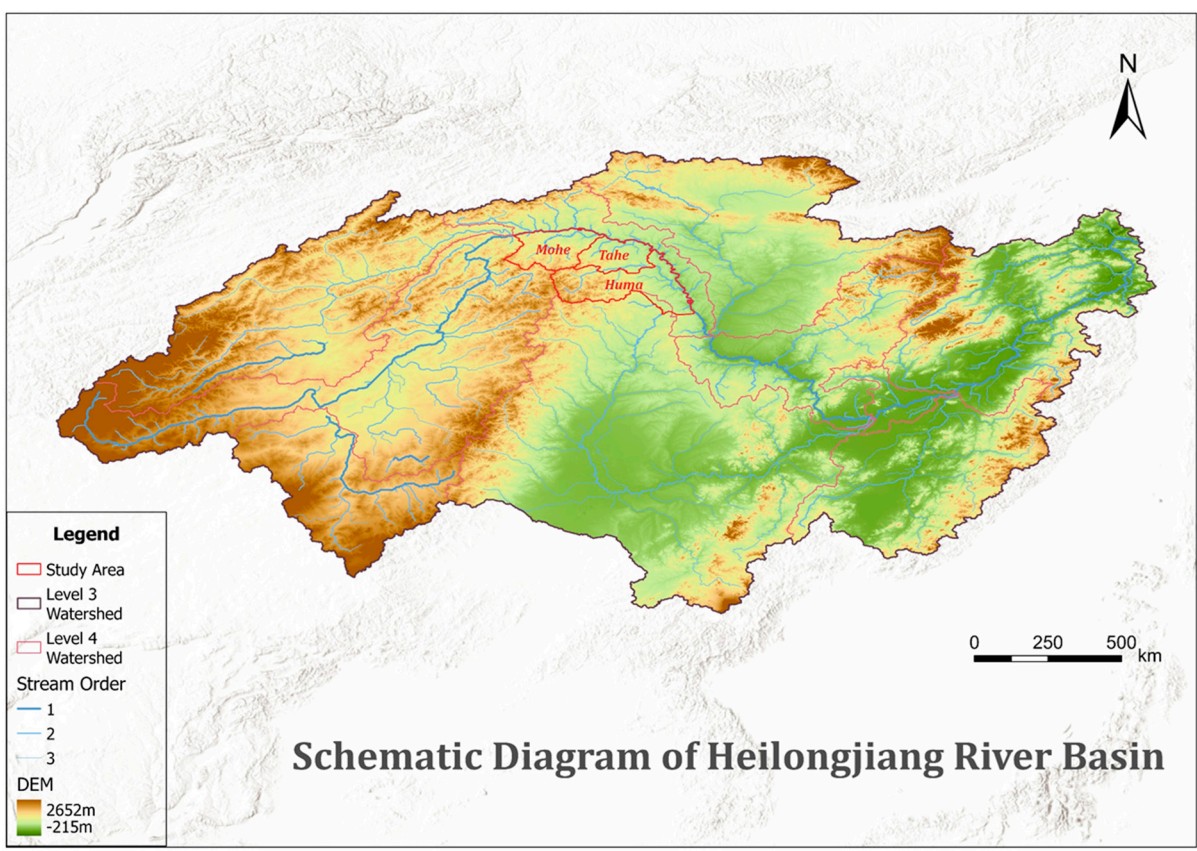

**Figure 2.** The geographic location of the study area.

### 3.2. Analysis of Ice Flood Risk

During the non-ice-blocked period of the ice-flowing period, the flow pattern of a river remains in a long-term linear relationship with hydraulic parameters such as channel characteristics, precipitation, snowmelt, and runoff. It exists in an equilibrium state. With changes in channel parameters, air temperature, and other conditions, ice floe may accumulate somewhere downstream of the channel or be pushed under the ice sheet, resulting in ice jams or dams. This disrupts the original balance of the water level in the river, resulting in a sudden change due to reduced flow velocity or backwater caused by ice blockages.

However, in the risk evaluation, it is also necessary to consider the differences in ice flood prevention and disaster resilience capacities in different regions. The above analysis indicates that the sudden accumulation of ice drains during the transition from stability to instability is the fundamental characteristic of potential disasters during the ice flood period. Therefore, it is possible to establish a risk evaluation index system and a model for evaluating the sudden changes in ice flood risks based on the catastrophe theory.

Given the challenges faced by certain methods, such as fuzzy evaluation methods which struggle with accurate weight determination and other issues involving factor analysis, complex calculations, and a substantial sample size requirement, opting for the catastrophe theory evaluation method is a reasonable choice to overcome these issues. The key advantage of this method is that it determines the importance of each indicator based on the inherent contradictions and mechanisms of various objectives within the normalization formula itself, without relying on indicator weights. As a result, the evaluation outcomes are objective in nature.

### 3.3. Data Acquisition and Processing

The research data used in this study were obtained from multiple sources, including the NOAA—National Centers for Environmental Information, ECMWF ERA5-Land monthly averaged data from 1981 to present, "China County Statistical Yearbook" (2000–2021), and the China Basic Geographic Information Sharing Website. The data underwent analysis and processing utilizing various tools such as ArcGIS Pro, SPSS 27.0, Origin 2022, and Excel 2021.

The risk of ice flood is influenced by multiple factors. This paper categorizes these factors into four guideline layers: hazard-inducing environment, hazard factors, hazard-bearing body, and anti-icing capability. Each subsystem plays a distinct role in the evolution of risk, with varying degrees of influence. Therefore, predicting the evolution of ice flood risk is a complex and uncertain task. To construct a risk evaluation index system that adheres to the principles of scientific, typical, comprehensive, systematic, and practical criteria, a four-level ice flood risk evaluation index system is developed. A total of 21 preliminary indicators are selected by integrating available information, as illustrated in Table 2.

**Table 2.** Preliminary selection of ice flood risk evaluation index system.

| Criterion Layer | Index Layer | Indicator Nature | Unit |
|---|---|---|---|
| Hazard-inducing Environment | River length ($X_{Q1}$) | (+) | km |
| | River gradient ($X_{Q2}$) | (+) | — |
| | Meander coefficient ($X_{Q3}$) | (+) | — |
| | Width-to-narrow ratio of sudden contraction in the river channel ($X_{Q4}$) | (+) | — |
| Hazard Factor | Upstream average temperature from October to March ($X_{P1}$) | (−) | °C |
| | Local average temperature from October to March ($X_{P2}$) | (−) | °C |
| | Upstream cumulative precipitation from November to March ($X_{P3}$) | (−) | mm |
| | Average temperature from April 1 to 20 ($X_{P4}$) | (+) | °C |
| | Average high temperature from April 1 to 20 ($X_{P5}$) | (+) | °C |
| | Upstream cumulative insolation from April 1 to 20 ($X_{P6}$) | (+) | h |
| | Local cumulative insolation from April 1 to 20 ($X_{P7}$) | (+) | h |
| | Snow depth on April 1($X_{P8}$) | (+) | mm |
| | Upstream average ice thickness in March ($X_{P9}$) | (+) | m |
| | Local average ice thickness in March ($X_{P10}$) | (+) | m |
| | Downstream average ice thickness in March ($X_{P11}$) | (+) | m |
| Hazard-bearing Body | Population density ($X_{R1}$) | (+) | people per km$^2$ |
| | Primary industry value-added ratio ($X_{R2}$) | (+) | — |
| | GDP per capita coefficient ($X_{R3}$) | (+) | — |
| Anti-icing Capability | Number of hospital beds per capita ($X_{S1}$) | (−) | sheet per people |
| | Resident deposit amount coefficient ($X_{S2}$) | (−) | — |
| | Local fiscal general budget revenue coefficient ($X_{S3}$) | (−) | — |
| Auxiliary Parameters | Ice flood hazard coefficient ($X_{M}$) | (+) | m·d |
| | Frequency of ice flood ($X_{N}$) | (+) | times |

Note: "+" in the table represents that the indicator promotes the development of disaster risk in the ice flood and "−" represents the suppression of the development of disaster risk in the ice flood.

## 4. Results and Discussion

### 4.1. Construct the Ice Flood Risk Evaluation Index System

The risk of an ice flood is influenced by numerous factors, and there exist complex non-linear relationships among these factors. When these factors undergo changes and interact with each other, the risk of an ice flood disaster can experience sudden variations. Figure 3a displays the correlation between hazard-inducing environmental factors and the frequency of ice floods in the Mohe, Tahe, and Huma regions. It indicates a low correlation between river length and the occurrence of ice floods. It is important to note that there is a strong negative correlation between $X_{Q3}$ and $X_N$, which is due to the small size of the dataset. Previous studies [9,44–46] have already shown a positive correlation between the river meander coefficient and the frequency of ice floods. The correlation between the hazard factor and the ice flood hazard coefficient is depicted in Figure 3b. As the significant level values for indicators $X_{P3}$, $X_{P6}$, and $X_{P7}$ are greater than 0.05, they are removed from the analysis. Additionally, there is a strong correlation between indicators $X_{P4}$ and $X_{P5}$, as well as $X_{P10}$ and $X_{P11}$, indicating redundancy in these indicators. Therefore, only $X_{P4}$ and $X_{P11}$, which exhibit a higher correlation with $X_M$, are retained.

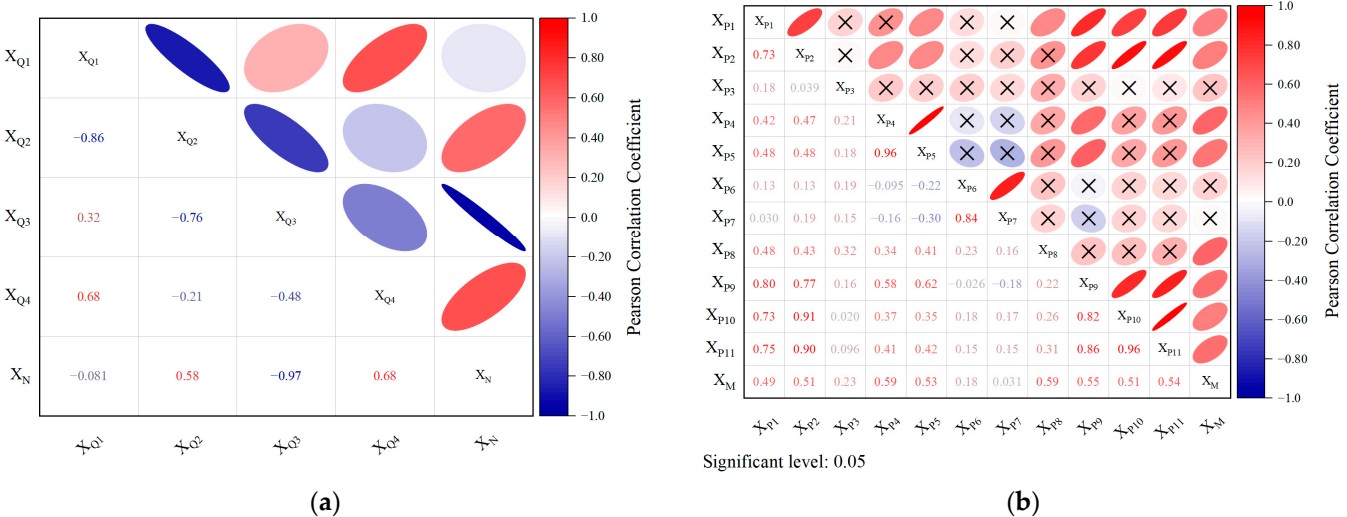

(**a**)                                                                 (**b**)

**Figure 3.** Pearson correlation analysis for preliminary selection of indicators. (**a**) The correlation between hazard-inducing environmental factors and the frequency of ice flood. (**b**) The correlation between the hazard factor and the ice flood hazard coefficient.

The filtered variables $X_{P1}$, $X_{P2}$, $X_{P4}$, $X_{P8}$, $X_{P9}$, and $X_{P11}$ undergo a hierarchical cluster analysis, resulting in the formation of the temperature element layer $C_4$ and the hydrological element layer $C_5$, as shown in Figure 4. It is important to highlight that when the clustering results of indicators lead to a distinct category, it signifies a crucial aspect of the evaluation system that was directly selected. Examples of such indicators include $X_{R1}$, $X_{R2}$, $X_{R3}$, $X_{S1}$, $X_{S2}$, and $X_{S3}$. These indicators hold significant value in the evaluation process.

Through the utilization of the Pearson correlation coefficient and hierarchical cluster analysis, a total of 15 indicators are selected as the final set of ice flood risk evaluation indicators. The priority ranking of the selected indicator set was determined based on previous studies [9], correlation coefficients, the vulnerability of the vulnerable entity, and the capacity for ice flood prevention. Please refer to Table 3 for more details on these indicators.

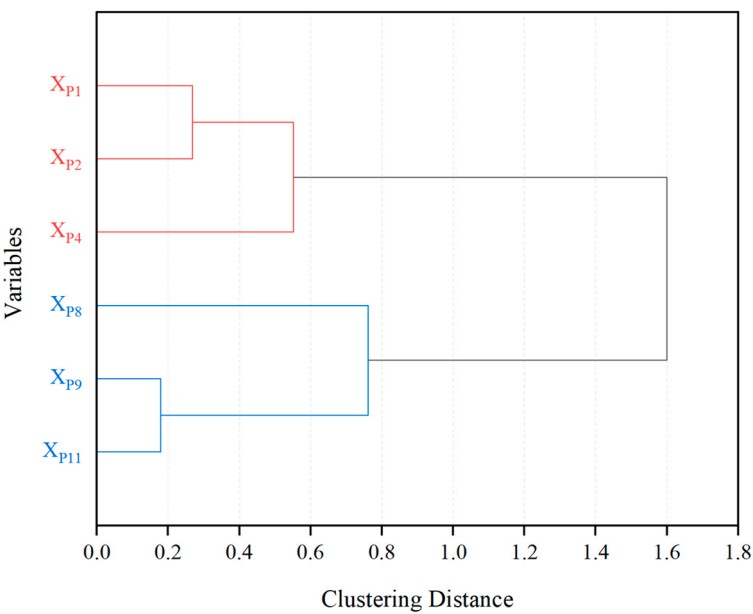

**Figure 4.** Hierarchical cluster analysis chart of hazard factors.

**Table 3.** Ice flood risk evaluation index system.

| Target Layer | Criterion Layer | | Index Layer | Pearson Correlation Coefficient | | Whether to Retain | Clustering Category | Indicator Layer |
|---|---|---|---|---|---|---|---|---|
| | | | | Correlation | Significant Level | | | |
| Comprehensive Risk Situation of Ice Flood ($A$) | Hazard-inducing Environment ($B_1$) | | $X_{Q1}$ | $-0.08$ | 0.949 | N | | |
| | | | $X_{Q2}$ | 0.58 | 0.609 | Y | 1 | $C_3$ |
| | | | $X_{Q3}$ | $-0.97$ | 0.154 | Y | 1 | $C_2$ |
| | | | $X_{Q4}$ | 0.68 | 0.526 | Y | 1 | $C_1$ |
| | Hazard Factor ($B_2$) | Climatic Elements ($C_4$) | $X_{P1}$ | 0.49 | 0.025 | Y | 2 | $D_3$ |
| | | | $X_{P2}$ | 0.51 | 0.019 | Y | 2 | $D_2$ |
| | | | $X_{P3}$ | 0.23 | 0.307 | N | | |
| | | | $X_{P4}$ | 0.59 | 0.005 | Y | 2 | $D_1$ |
| | | | $X_{P5}$ | 0.53 | 0.013 | N | | |
| | | | $X_{P6}$ | 0.18 | 0.438 | N | | |
| | | | $X_{P7}$ | 0.03 | 0.893 | N | | |
| | | Hydrological Elements ($C_5$) | $X_{P8}$ | 0.59 | 0.005 | Y | 3 | $D_4$ |
| | | | $X_{P9}$ | 0.55 | 0.010 | Y | 3 | $D_5$ |
| | | | $X_{P10}$ | 0.51 | 0.018 | N | | |
| | | | $X_{P11}$ | 0.54 | 0.011 | Y | 3 | $D_6$ |
| | Hazard-bearing Body ($B_3$) | | $X_{R1}$ | | | Y | 4 | $C_6$ |
| | | | $X_{R2}$ | | | Y | 4 | $C_7$ |
| | | | $X_{R3}$ | | | Y | 4 | $C_8$ |
| | Anti-icing Capability ($B_4$) | | $X_{S1}$ | | | Y | 5 | $C_9$ |
| | | | $X_{S2}$ | | | Y | 5 | $C_{10}$ |
| | | | $X_{S3}$ | | | Y | 5 | $C_{11}$ |

Therefore, a four-level evaluation index system is established. The first layer is the target layer $A$, which is the evaluation of ice flood risk; the second layer is the criterion layer $B$, which is the hazard-inducing environment $B_1$, the hazard factor $B_2$, the hazard-bearing body $B_3$, and the anti-icing capability $B_4$; the third layer $C$ is the indicators $C_1$ to $C_{11}$ obtained after screening; and $C_4$ and $C_5$ consist of the bottom indicators $D_1$ to $D_6$, representing the influence of climatic elements and hydrological elements, as shown in Table 3.

*4.2. Ice Flood Risk Evaluation Results and Grade Classification*

In the criterion layer, the hazard-inducing environment, climatic elements, hydrological elements, hazard-bearing body, and anti-icing capability ($B_1$, $C_4$, $C_5$, $B_3$, and $B_4$) are composed of three variables, following a swallowtail catastrophe model. In addition, the hazard factor ($B_2$) in the criterion layer consists of two variables, following a sharp point catastrophe model. In the target layer, the ice flood hazard risk ($A$) is composed of four variables ($B_1$, $B_2$, $B_3$, and $B_4$), which follow a butterfly catastrophe model.

The data from the indicator layer of Mohe, Tahe, and Huma from 2000 to 2020 were incorporated into the mutation evaluation method. Over the 20-year period, ice dams and jams occurred in the upper Heilongjiang River during the ten years. The results of the ice flood risk evaluation and the classification of each region in the study year can be observed in Figure 5.

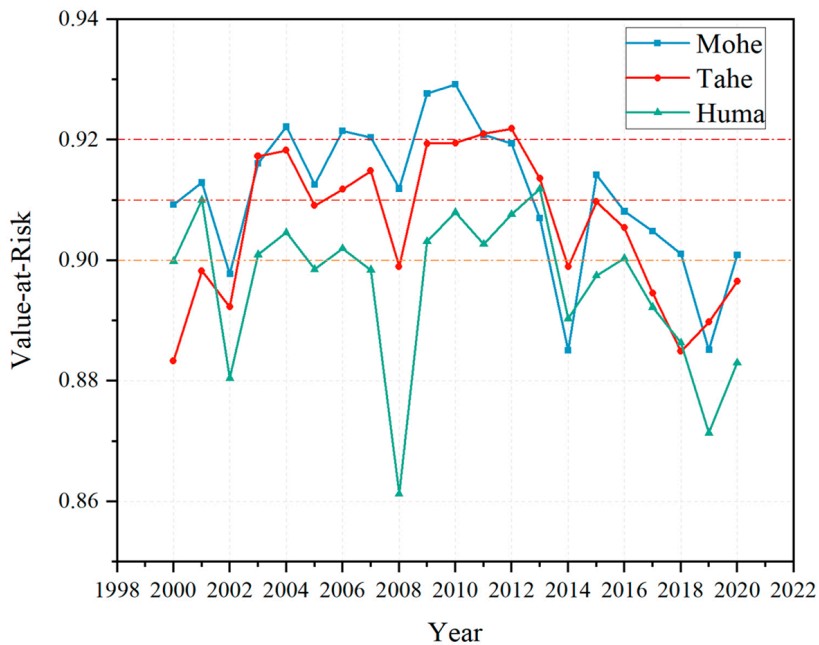

**Figure 5.** The results and classification of ice flood risk evaluation in the study year.

At present, there is no unified standard for the evaluation of ice flood risk, either domestically or internationally. Referring to the relevant studies [47,48], the evaluation criteria of their ice flood hazard evaluation index were determined by combining the actual situations of ice dam flooding in the upper reaches of the Heilongjiang River. Performing a hierarchical cluster analysis on the set of indicators for each year, as shown in Figure 6, we selected the years 2000, 2001, 2008, 2009, 2010, 2011, and 2015 as representative years based on the results of the clustering analysis. The clustered distances, value-at-risk, and realistic risk ratings for the typical years in the three regions are shown in Table 4.

The ice flood risk for each year and region was ranked according to the results of the catastrophe theory evaluation. Based on the historical ice flood data for the upstream region of the Heilongjiang River during the representative years, the evaluation results were divided into different levels. By following this approach, we determined the grade intervals corresponding to the risk levels of ice floods in the upper reaches of Heilongjiang Province. The specific grade intervals and corresponding details can be found in Table 5.

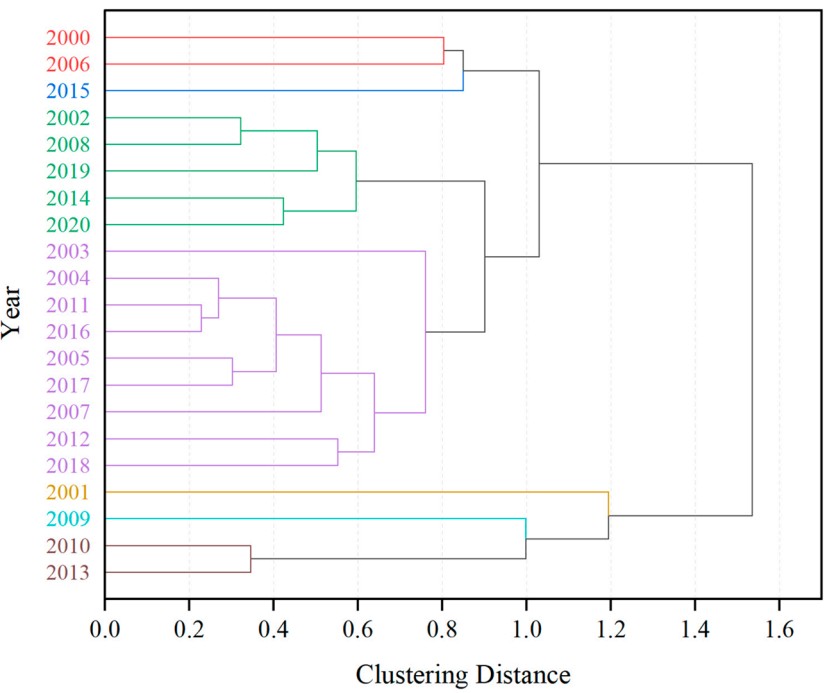

**Figure 6.** Clustering results of index sets for each year.

**Table 4.** The clustered distances, value-at-risk, and realistic risk ratings for the typical years.

| Clustering Group | Year | Clustering Distance | Mohe | | Tahe | | Huma | |
|---|---|---|---|---|---|---|---|---|
| | | | Value | Rating | Value | Rating | Value | Rating |
| 1 | 2000 | 0.804 | 0.909 | Moderate | 0.883 | Low | 0.900 | Moderate |
| 2 | 2015 | 0.850 | 0.914 | High | 0.910 | High | 0.897 | Low |
| 3 | 2008 | 0.322 | 0.912 | High | 0.899 | Low | 0.861 | Low |
| 4 | 2011 | 0.229 | 0.921 | Critical | 0.921 | Critical | 0.903 | Moderate |
| 5 | 2001 | 1.195 | 0.913 | High | 0.898 | Low | 0.910 | High |
| 6 | 2009 | 0.998 | 0.928 | Critical | 0.919 | High | 0.903 | Moderate |
| 7 | 2010 | 0.346 | 0.929 | Critical | 0.919 | High | 0.908 | Moderate |

Note: Selection method: sum of distances.

**Table 5.** The classification of ice flood risk.

| Typical Years and Regions | | Evaluation Results (*A*) | Grading Results | |
|---|---|---|---|---|
| Year | Region | | Range | Grade |
| 2010 | Mohe | 0.929 | 0.92~1 | I |
| 2011 | Mohe | 0.921 | | |
| 2010 | Tahe | 0.920 | 0.91~0.92 | II |
| 2001 | Huma | 0.910 | | |
| 2000 | Mohe | 0.909 | 0.90~0.91 | III |
| 2000 | Huma | 0.900 | | |
| 2008 | Tahe | 0.899 | 0~0.90 | IV |
| 2008 | Huma | 0.861 | | |

*4.3. Results Analysis*

A comparison of the average risk values in Mohe, Tahe, and Huma from a geographical perspective indicates that the evaluation level of ice flood risk in the upper main stream of the Heilongjiang River was highest in Mohe, followed by Tahe, and then Huma, throughout the period from 2000 to 2020. From the perspective of the hazard-inducing environment,

the Mohe segment is characterized by the maximum river gradient and width-to-narrow ratio of sudden contraction in the river channel. The ratio compares the width of the original river section to the width of a narrowed section. A larger width-to-narrow ratio implies a more significant change in the width of the river channel. When a significant amount of ice flows through this section, the width of the channel decreases sharply, and ice debris is more likely to accumulate and cause congestion. As a result, the hazard risk of ice flood is higher in this particular segment of the river.

In addition, the Huma segment of the river gradient has a larger meander coefficient, indicating a steeper course compared to other sections. The river gradient is calculated as the ratio of the height difference between starting and ending points to the actual length of the river. Consequently, ice slush tends to accumulate in these bends, obstructing the water flow and quickly raising water levels. Together, the combination of a larger river gradient and a higher width-to-narrow ratio in the Mohe section increases the risk of ice floe accumulation, water flow blockage, and the subsequent occurrence of ice dams or jams.

From a temporal perspective, the risk levels in the Mohe, Tahe, and Huma areas follow a pattern of initial increase followed by a subsequent decrease. There was a notable increase in the overall risk index from 2000 to 2010. All three regions increased to the higher risk level, with the Tahe region showing the greatest increase at 4%. Regarding the factors contributing to these changes, the average temperature upstream during the October 2009 to March 2010 period was 3.1 °C higher compared to 2000. This period corresponds to the freeze-up phase of the Heilongjiang River, and the average temperature during this time directly affects the volume of ice and water in the river during the opening period. Furthermore, the local fiscal general budget revenue coefficient and the coefficient of resident savings deposits are generally lower compared to the year 2000. Additionally, the proportion of agricultural output value is significant. These factors contribute to an increased potential risk for the occurrence of ice floods.

In comparison to 2009, the risk levels of the three regions escalated in 2010. This can be attributed to the lower average temperatures experienced along the Heilongjiang River in April, with Mohe region being 4.68 °C lower than previous years, and Huma maintaining temperatures below zero. Temperature serves as a critical thermal condition for ice flood formation. In 2010, the Mohe and Huma sections failed to thaw due to the persistently low average temperatures. As a result, the upstream water carrying a substantial amount of floating ice exerted pressure on the downstream ice cover, leading to ice squeezing and accumulating. This scenario created favorable conditions for the formation of ice dams and jams.

The Heilongjiang River is primarily lined with villages, and a majority of rural residents rely on agriculture as their main source of livelihood. Consequently, during ice flood disasters, agriculture, in addition to the population, becomes the primary vulnerable entity. The severity of the consequences resulting from an ice flood disaster is directly proportional to the population density and per capita food possession. Enhancing the number of drainage structures and extending the length of embankments will enhance the region's capacity to mitigate ice flood disasters. As the proportion of total agricultural output in the regional GDP increases, the recovery process from the impact of ice flood disasters becomes more challenging. On the whole, the risk level of ice floods in the upper reaches of the Heilongjiang River demonstrates a decreasing trend, indicating a yearly improvement in the economic level and the ice flood prevention capabilities in the region.

*4.4. Accuracy Evaluation*

In order to examine the accuracy of the catastrophe evaluation method, the entropy weighting method was used to assign weights to the underlying indicators in the index system. The fuzzy comprehensive evaluation method was used to evaluate the risk of ice floods in Mohe, Tahe, and Huma regions from 2000 to 2020. Regression analysis of the underlying indicators and evaluation results showed that the correlation coefficient ($R^2$) based on the catastrophe evaluation method was 0.997 and the root mean square error

(*RMSE*) was 0.00898; the correlation coefficient ($R^2$) based on the fuzzy evaluation method was 0.995 and the root mean square error (*RMSE*) was 0.00002. The root mean square error of both algorithms is less than 0.05, and the correlation coefficients are significantly correlated, indicating that the catastrophe theory evaluation method has high accuracy in the application of ice flood hazard evaluation. The average risk values obtained from the fuzzy comprehensive evaluation method and the catastrophe theory evaluation method were compared for the Mohe region as presented in Figure 7 and Table 6. The results indicate that the two methods yielded relatively similar results, and the levels of risk values matched perfectly.

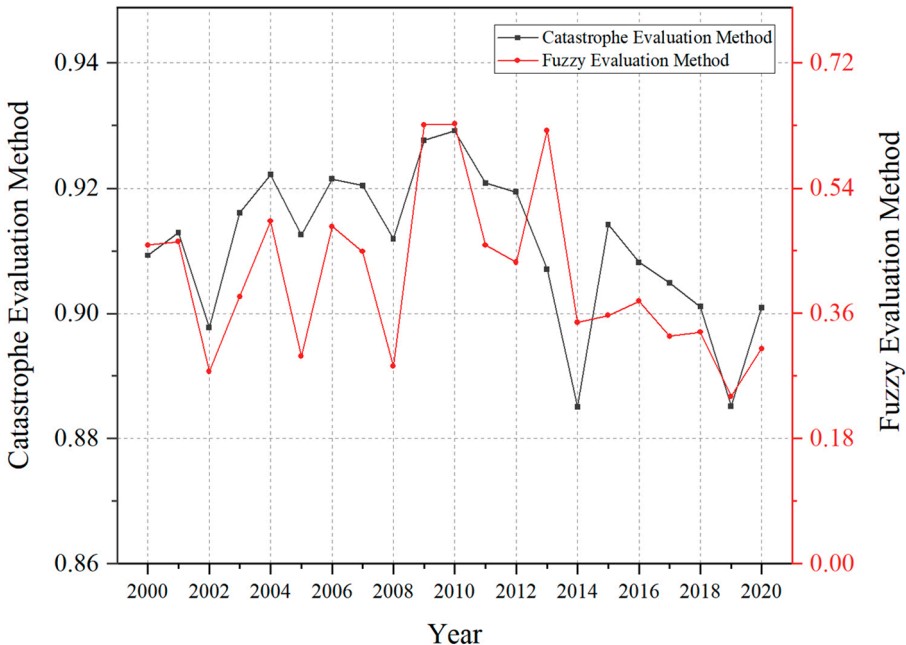

**Figure 7.** Comparison of the results of the ice flood risk evaluation.

**Table 6.** The level of value-at-risk for typical years.

| Year | Catastrophe Evaluation Method | | Fuzzy Evaluation Method | |
|---|---|---|---|---|
| | **Value-at-Risk** | **Level** | **Value-at-Risk** | **Level** |
| 2010 | 0.929 | 1 | 0.632 | 1 |
| 2009 | 0.928 | 2 | 0.631 | 2 |
| 2011 | 0.921 | 3 | 0.458 | 3 |
| 2015 | 0.914 | 4 | 0.357 | 6 |
| 2001 | 0.913 | 5 | 0.463 | 4 |
| 2008 | 0.912 | 6 | 0.284 | 7 |
| 2000 | 0.909 | 7 | 0.458 | 5 |

## 5. Conclusions and Future Prospective

This study employed the catastrophe evaluation method to evaluate the risk of ice dam floods in the upper Heilongjiang River spanning from 2000 to 2020. The evaluation findings indicated that the Mohe section, characterized by an intricate and steep river topography, exhibits a higher comprehensive ranking of ice flood risk compared to the other two regions. Regarding the time series analysis, ice floods tend to occur more frequently during years with lower upstream temperatures between October and March, coupled with larger upstream and downstream ice thickness in March. Population, agriculture, economy, and other factors also affect the risk of the occurrence of floods, resulting in a trend where ice flood risk initially increases and then decreases. The results obtained through the catastrophe evaluation method exhibit a similar risk ranking as the fuzzy evaluation method. Furthermore, the catastrophe evaluation method offers the advantages

of simplicity in calculation and reduced subjective factors. It eliminates the requirement for precise weighting of underlying indicators and results in a more rational overall distribution of risk values.

In addition, this study still has some limitations that need to be addressed and explored in future research:

1. Problems such as insufficient selection of indicators due to the difficulty of data accessibility may have some influence on the results of the ice flood disaster risk evaluation. However, as the construction and enhancement of the big data platform progress, it will be possible to include a wider range of indicators to enhance the ice flood disaster risk evaluation system. This improvement will contribute to more accurate and reliable results in the future.

2. Using the entropy weight method, in the fuzzy comprehensive evaluation method, to determine the weight of the index may result in distorted evaluation outcomes due to inaccuracies in some of the weights. In future research, we plan to explore alternative weighting techniques or enhanced fuzzy theory to obtain more robust and desirable conclusions. By doing so, we aim to address the limitations and potential distortions associated with the entropy weight method and improve the overall accuracy and reliability of our evaluations.

**Author Contributions:** Conceptualization, H.H. and E.W.; methodology, Y.L., H.H. and X.L.; formal analysis, H.H. and E.W.; investigation, Y.L., H.H., Y.S., X.X. and H.L.; data curation, Y.L., Y.S., X.X. and H.L.; writing—original draft preparation, Y.L. and H.H.; writing—review and editing, Y.L. and X.L.; funding acquisition, H.H. All authors have read and agreed to the published version of the manuscript.

**Funding:** This research was supported by the Major Scientific and Technological Projects of the Ministry of Water Resources of China (No. SKS-2022017), the Natural Science Foundation of Heilongjiang Province of China (No. LH2020E004), and the Project to Support the Development of Young Talent by Northeast Agricultural University.

**Data Availability Statement:** The data are available upon request.

**Conflicts of Interest:** The authors declare no conflict of interest.

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
