# Peer review of "Risk Evaluation of Ice Flood Disaster in the Upper Heilongjiang River Based on Catastrophe Theory"

_water, doi:10.3390/w15152724_

Round 1
Reviewer 1 Report
Title
The title of the manuscript is OK.
Graphical abstract & abstract
The manuscript's abstract needs a bit clarification (make it more concise, it seems like basic information was given while important and major parts are missing that needs to be addressed in a better way). Moreover, the graphical abstract should be added for the enhancement of the readings and citations but it is not necessary.
Introduction
The author of the manuscript manage the references in a proper. The style of references is ok and adjusted in a proper sequence, while some material in the introduction is required. The author of the manuscript can take help using the below DOIs that play a significant role in the enhancement of the quality of the introduction.
- 10.1007/s12633-022-01878-2
ii) 10.1142/S0217979224500553
iii) 10.1515/zpch-2023-0206
If the author used these DOIs in the manuscript, than follow the above-mentioned instructions and ignore the DOIs.
Results and Discussion
The elaboration and description between various relationships was managed in a sequential manner but in the figures 3(a and b) the written terms and not seen clearly, to make them visible try to use different colors while the data of all the tables were managed in a significant way.
Material and method
From the line, 122 to 124 the fold catastrophe type, cusp catastrophe type, swallowtail catastrophe type, and butterfly catastrophe type are commonly used primary catastrophe models was written it should be changed to different sorts of primary catastrophe models are used which includes 124 the fold catastrophe type, cusp catastrophe type, swallowtail catastrophe type, and butterfly catastrophe type etc. from the heading 2.1 to onwards subheadings should be arranged e.g. 2.1.1 , 2.2.1 etc.
The line 170 The Pearson correlation coefficient, often referred to as Pearson's r or simply should be corrected as The Pearson correlation coefficient, often referred to as Pearson's or simply, while At some places grammatical errors were found so, this should be reduced using Grammarly.
Conclusion
Write the conclusion in paragraph form and add future prospects in numbered form.
NA
Reviewer 2 Report
MS on 'Risk Evaluation of Ice Flood Disaster in the Upper Hei-longjiang River Based on Catastrophe Theory' presents a case study on ice flood disaster risk assessment as applicable for frigid locations of high latitude and high altitude. Some comments are as follows:
1. Kindly mention the elevation unit in figure 2. Kindly check the lowest value of DEM given as -215. Also, it is difficult to differentiate between level 3 and 4 watershed.
2. How the indicator layers were prioritised as given in Table 3 (Ice flood risk evaluation index system). What is/ are the criteria for layer retention?
3. A systematic lag for some years between three sites are observed in figure 5 (The results and classification of ice flood risk evaluation in the study year). Please elaborate.
4. How the specific representative years were selected (2000, 2001, 2008, 2010, 2011 and 2016) based on clustering analysis?
5. How the fuzzy evaluation method was incorporated in the study is not clear.
OK.
Reviewer 3 Report
- Major English language issues
- Abstract should include the major results of the study
- Line 16- should be This Study
- Line 40 – put a dot before thus and start a new sentence
- Line 82 – add 25 near Wu
- Line 91- This study
- Line 233 – cut refer to and leave just Figure 2
- Figure 2 – delete Schematic diagram
- Figure 3 should be more visible and clear
- Subheading 2.2. Pearson Correlation Coefficient – should be decribed more concisely and do not present in very detailed manner
- 2.3. Hierarchical Cluster Analysis – what similarity/dissimilarity index was used to group the variables?
- The data should be presented in a logical, coeherent and structured manner. The structure of the manuscript it’s hard to follow in the current state.
- Major English language issues
